

# Impact of non-ideality on reconstructing spatial and temporal variations of aerosol acidity with multiphase buffer theory

Guangjie Zheng[1], Hang Su[2], Siwen Wang[2], Andrea Pozzer[3], Yafang Cheng[1*]

[1] Minerva Research Group, Max Planck Institute for Chemistry, Mainz 55128, Germany.
[2] Multiphase Chemistry Department, Max Planck Institute for Chemistry, Mainz 55128, Germany.
[3] Air Chemistry Department, Max Planck Institute for Chemistry, Mainz 55128, Germany.

*Correspondence to*: Y. Cheng (yafang.cheng@mpic.de)

**Abstract.** Aerosol acidity is a key parameter in atmospheric aqueous chemistry and strongly influence the interactions of air pollutants and ecosystem. The recently proposed multiphase buffer theory provides a framework to reconstruct long-term trends and spatial variations of aerosol pH based on the effective acid dissociation constant of ammonia ($K_{a,NH3}^*$). However, non-ideality in aerosol droplets is a major challenge limiting its broad applications. Here, we introduced a non-ideality correction factor ($c_{ni}$) and investigated its governing factors. We found that besides relative humidity (RH) and temperature,

$c_{ni}$ is mainly determined by the molar fraction of $NO_3^-$ in aqueous-phase anions, due to different $NH_4^+$ activity coefficients between $(NH_4)_2SO_4$- and $NH_4NO_3$-dominated aerosols. A parameterization method is thus proposed to estimate $c_{ni}$ at given RH, temperature and $NO_3^-$ fraction, and is validated against long-term observations and global simulations. In the ammonia-buffered regime, with $c_{ni}$ correction the buffer theory can well reproduce the $K_{a,NH3}^*$ predicted by comprehensive thermodynamic models, with root-mean-square deviation ~0.1 and correlation coefficient ~1. Note that, while $c_{ni}$ is needed to

predict $K_{a,NH3}^*$ levels, it is usually not the dominant contributor to its variations, as ~90% of the temporal or spatial variations in $K_{a,NH3}^*$ is due to variations in aerosol water and temperature.



## 1. Introduction

Aerosol acidity strongly influences the thermodynamics and chemical kinetics of atmospheric aerosols and is therefore one
essential parameter in evaluating their environmental, health and climate effects (Pye et al., 2020; Zheng et al., 2020). However, direct measurements of aerosol pH in the real atmosphere are not available so far (Pye et al., 2020; Li et al., 2020). The fast equilibrium with ambient air, tiny volume and high ionic strength and nucleation potential are the main challenges for measurements, especially online or in-situ measurements. Several groups are developing new techniques for this purpose (Wei et al., 2018; Craig et al., 2018; Li et al., 2020; Ault, 2020). For example, Wei et al. (2018) developed an in-situ Raman
microscopy method for pH measurements in microdroplets (diameter ~20 $\mu$m), with an uncertainty of ~0.5 pH units. Craig et al. (2018) and Li et al. (2020) developed colorimetric analyses on pH-indicator papers for aerosol pH measurement, which exhibit uncertainties around 0.4-0.5 pH units. These currently available techniques, however, still need to be developed further for real atmospheric applications.

Due to the lack of direct measurements, modelling tools have been intensively used to calculate the aerosol pH (Fountoukis
and Nenes, 2007; Fountoukis et al., 2009; Clegg et al., 2001; Zuend et al., 2008). Results of thermodynamic models are subject to uncertainties in the input parameters (Fountoukis et al., 2009; Pye et al., 2020; Weber et al., 2016; Guo et al., 2016; Guo et al., 2017; Pye et al., 2018; Tao and Murphy, 2019; Hennigan et al., 2015; Guo et al., 2015; Peng et al., 2019). For example, Hennigan et al. (2015) revealed the importance of including gas-phase species in the input, in addition to the full aerosol composition measurements (Fountoukis et al., 2009). Guo et al. (2015) suggests overall uncertainties of ~0.2-0.5 pH units
related to aerosol composition. Pye et al. (2020) reviewed major thermodynamic models and show that the estimated acidity among different models were on average 0.3 pH units, but sometimes as much as 1 pH unit.

The recently proposed multiphase buffer theory shows that globally most of the populated urban areas are within the multiphase ammonia-buffered regime (Zheng et al., 2020). In the buffered regions/periods, $pK_{a,NH3}^{*}$ can serve as a proxy of aerosol pH, where $K_{a,NH3}^{*}$ is the effective acid dissociation constant of $NH_3$ in multiphase systems (section 2). Ideally, $pK_{a,NH3}^{*}$ is fully
determined by aerosol water content (AWC) and temperature. However, the non-ideality in aerosols may introduce deviations from the ideal conditions. Here we investigated such deviation and derived a non-ideality correction factor for using $pK_{a,NH3}^{*}$ as a proxy of aerosol pH. Governing factors of the non-ideality correction factor in aerosol droplet are further explored and discussed, based on which a parameterization method to estimate the non-ideality correction factors are proposed. We also estimated that a constant correction factor of $pK_{a,NH3}^{*}$ is often good enough to predict pH over a period at a given site, or to
explain the global pH variations. We thereby provided a way for pH retrieval when chemical measurements are unavailable for the ammonia-buffered regions and periods.



## 2. Methods

### 2.1 Effective acid dissociation constant as a proxy of aerosol pH

**Acid dissociation constant of NH₃ in bulk solutions, $K_a$**

The definition of acids and bases have been evolving over time (Zheng et al., 2020). The pioneering Arrhenius theory defined base as a substance that dissociates in water to form hydroxide (OH⁻) ions (Pfennig, 2015). Therefore, an Arrhenius base can be expressed as $B_AOH$, which dissociation in water as:

$$B_AOH \rightleftharpoons B_A^+ + OH^- \tag{1a}$$

with the corresponding base dissociation constant $K_b$ being:

$$K_b = [B_A^+]\,[OH^-]\,/\,[B_AOH] \tag{1b}$$

In combination with the water dissociation of:

$$H_2O \rightleftharpoons H^+ + OH^-, \qquad K_w = [H^+]\,[OH^-] \tag{2}$$

The corresponding acid dissociation constant, defined as $K_a = K_w\,/\,K_b$, is thus (reaction 2 – reaction 1):

$$B_A^+ + H_2O \rightleftharpoons B_AOH + H^+, \qquad K_a = [B_AOH]\,[H^+]\,/\,[B_A^+] \tag{3}$$

The later Bronsted-Lowry theory defined base as proton acceptor (Pfennig, 2015), and is expressed as $B_{BL}$ here. In this sense, an Arrhenius base $B_AOH$ is not considered as a Brønsted base, but rather salts. The dissociation reaction for a Brønsted base is expressed as:

$$B_{BL} + H_2O \rightleftharpoons B_{BL}H^+ + OH^-, \qquad K_b = [B_{BL}H^+]\,[OH^-]\,/\,[B_{BL}] \tag{4}$$

and the corresponding $K_a$ is thus (reaction 2 – reaction 4):

$$B_{BL}H^+ \rightleftharpoons B_{BL} + H^+, \qquad K_a = [B_{BL}]\,[H^+]\,/\,[B_{BL}H^+] \tag{5}$$

As NH₃(aq) is actually the water adduct of NH₃, it is often be expressed equivalently as NH₃(aq) = NH₃·H₂O(aq) = NH₄OH(aq). In this sense, it can fit in the category of both definitions. In the Arrhenius definition, the base $B_AOH$ = NH₄OH, namely $B_A$= NH₄⁺. Therefore, $K_{a,\,NH3}$ is (Eq. 3):

$$K_{a,NH3} = [NH_4OH(aq)]\,[H^+]\,/\,[\,NH_4^+] = [NH_3(aq)]\,[H^+]\,/\,[\,NH_4^+] \tag{6}$$

While with the Bronsted-Lowry definition, the base is $B_{BL}$ = NH₃(aq), and $K_{a,\,NH3}$ is (Eq. 5):

$$K_{a,NH3} = [NH_3(aq)]\,[H^+]\,/\,[\,NH_4^+] \tag{7}$$

which is the same as Eq. 6. Therefore, different definition of bases for the ammonia family ($B_A$= NH₄⁺ or $B_{BL}$ = NH₃(aq)) will led to the same expression of $K_{a,NH3}$, as defined in Zheng et al. (2020). The same applies for other volatile weak bases.





**Ideal multiphase acid dissociation constant of NH₃**

The multiphase effective acid dissociation constant of $NH_3$ under ideal conditions, $K_{a,NH3}^{*,i}$, depends only on AWC and temperature as (Zheng et al., 2020):

$$K_{a,NH3}^{*,i} = \frac{[H^+(aq)]([NH_3(aq)]+[NH_3(g)])}{[NH_4^+(aq)]} = K_{a,NH3}(1+\frac{\rho_w}{H_{NH3}\ R\ T\ AWC})$$

where AWC is mainly determined by air particulate matter concentrations and RH. The [$NH_3$ (g)] represents equivalent molality of gaseous $NH_3$ in solution (see details in Zheng et al. (2020)). The $H_{NH3}$ is Henry's law constant of $NH_3$ in mol L⁻¹

atm⁻¹, $R$ is the gas constant of 0.08205 atm L mol⁻¹ K⁻¹, $T$ is temperature in K, AWC is in $\mu g$ m⁻³, and $\rho_w$ is water density in $\mu g$ m⁻³.

For typical ambient conditions when AWC varies between 1 to 1000 $\mu g$ m⁻³, the [$NH_3$(g)] is usually $10^5$ to $10^8$ times larger than [$NH_3$(aq)], and the above equation can be simplified into:

$$K_{a,NH3}^{*,i} = \frac{[H^+(aq)][NH_3(g)]}{[NH_4^+(aq)]} = K_{a,NH3}\frac{\rho_w}{H_{NH3}\ R\ T\ AWC} \tag{8a}$$

And taking negative lognormal on both sides, we have pH is related to $pK_{a,NH3}^{*,i}$ (i.e., $-\log K_{a,NH3}^{*,i}$) as (Zheng et al., 2020):

$$pH = pK_{a,NH3}^{*,i}+\log\frac{[NH_3(g)]}{[NH_4^+(aq)]} \tag{8b}$$

**2.2 Influences of non-ideality on aerosol pH**

For ambient aerosols, the ionic strength ($I$) is high, and the non-ideality must be considered. Under such non-ideal conditions, the multiphase equilibrium of $NH_3$ can be expressed as (Zheng et al., 2020):


$$K_{a,NH3}^{*,i} = \frac{(\gamma_{NH3(g)}[NH_3(g)] + \gamma_{NH3(aq)}[NH_3(aq)])(\gamma_{H+}[H^+(aq)])}{\gamma_{NH4+}[NH_4^+(aq)]} \tag{9}$$

where $\gamma_X$ is the activity coefficient for species X.

Activity coefficients for gases, like $\gamma_{NH3(g)}$, are usually treated as unity. Again, for typical ambient conditions [$NH_3$(g)] is much larger than [$NH_3$(aq)], and $\gamma_{NH3(aq)}$[$NH_3$(aq)] can be omitted. Eq. 9 can thus be simplified into:

$$K_{a,NH3}^{*,i} = \frac{\gamma_{H+}}{\gamma_{NH4+}}\frac{[H^+(aq)][NH_3(g)]}{[NH_4^+(aq)]} \tag{10}$$

Under non-ideal conditions, pH is usually defined by the proton activity, i.e.:

$$pH_a = - \log (\gamma_{H+}[H^+]) \tag{11}$$

However, in thermodynamic models that are most commonly applied in current global models (ISORROPIA II, MOSAIC, etc.), the pH is usually defined as free-$H^+$ molality (Pye et al., 2020), i.e.:

$$pH = pH_F = - \log ([H^+]) \tag{12}$$





The difference of activity- and molality-defined pH (i.e., $pH_a$ and $pH_F$) is discussed in a previous study (Pye et al., 2020), which show that deviations of $pH_F$ from $pH_a$ is larger at lower RH, and is usually within 1 unit when RH > 60% (Pye et al., 2020). To be comparable with results in previous studies, the pH we discussed hereinafter follow the free-$H^+$ molality definition. Discussion based on activity-defined pH is detailed in Appendices A and B.

With the free-$H^+$ molality pH definition, multiphase buffer theory can be rewritten as (Eqs. 10 and 12):


$$K_{a,NH3}^{*,ni} = \frac{[NH_3(g)][H^+(aq)]}{[NH_4^+(aq)]} = \frac{\gamma_{NH4+}}{\gamma_{H+}} K_{a,NH3}^{*,i} = \frac{\gamma_{NH4+}}{\gamma_{H+}} K_{a,NH3} \frac{\rho_w}{H_{NH3} \, R \, T \, AWC}$$ (13a)

$$pH = -log([H^+(aq)]) = pK_{a,NH3}^{*,ni} + log \frac{[NH_3(g)]}{[NH_4^+(aq)]}$$ (13b)

where $K_a^{*,ni}$ is the multiphase effective acid dissociation constant under non-ideal conditions. The difference of pH caused by non-ideality is therefore (Eqs. 8 and 13):

$$c_{ni} = pK_a^{*,ni} - pK_a^{*,i} = -log \frac{\gamma_{NH4+}}{\gamma_{H+}}$$ (14a)

where $c_{ni}$ is hereinafter denoted as the non-ideality correction factor.

Another way to calculate $c_{ni}$ is by definition, i.e.:

$$c_{ni} = pK_a^{*,ni} - pK_a^{*,i} = -log(\frac{[NH_3(g)][H^+(aq)]}{[NH_4^+(aq)]}) + log(K_{a,NH3} \frac{\rho_w}{H_{NH3} \, R \, T \, AWC})$$ (14b)

For some thermodynamics models that predict both the activity coefficients of ions and the gas-particle partitioning of species like the E-AIM model (section 2.3), $c_{ni}$ can be derived either from Eq. 14a (activity-based) or Eq. 14b (gas-particle partitioning

based). However, current atmospheric chemical transport models usually adopted the more computation-efficient thermodynamic models (ISORROPIA II, MOSAIC, etc.), in which only the mean activity coefficient of an electrolyte species $ij$ in water, $\gamma_{ij}$, are derived, where $i$ is a cation while $j$ is an anion (Pye et al., 2020; Fountoukis and Nenes, 2007; Zaveri et al., 2005). For these models, we cannot directly derive $\gamma_{NH4+}$ or $\gamma_{H+}$, and $c_{ni}$ are derived through Eq. 14b (i.e., from the predicted $[NH_3]$, $[NH_4^+]$, $[H^+]$, and AWC).

**2.3 Model simulations**

**Thermodynamic models.** Here we used E-AIM model (model IV; http://www.aim.env.uea.ac.uk/aim/aim.php) (Clegg et al., 1992b; Wexler and Clegg, 2002; Friese and Ebel, 2010) to predict both the activity coefficients for individual ions and the gas-particle partitioning. The E-AIM model adopted the Pitzer-Simonson-Clegg model (Clegg et al., 1992a; Clegg et al., 1998) to calculate single-ion activity coefficients, which included most comprehensive conditions and have been used as a benchmark

(Clegg et al., 1992b; Hennigan et al., 2015; Pye et al., 2020). Therefore, both the activity-based pH ($pH_a$, Eq. 11) and the free-$H^+$ molality pH ($pH_F$, Eq. 12) can be derived (Appendix B). In addition, we also adopted the ISORROPIA v2.3 model





(Fountoukis and Nenes, 2007) for comparison, which is computational effective and has been commonly adopted in global and regional models. To reduce the computational cost, the ISORROPIA model calculated only the binary activity coefficients $\gamma_{ij}$ using the Kusik-Meissner relationship and the Bromley's formula (Fountoukis and Nenes, 2007). Therefore, only the free-

H$^+$ molality pH (pH$_F$, Eq. 12) can be derived in ISORROPIA (Appendix D). For example, for a HCl droplet, both the $\gamma_{H+(aq)}$

and $\gamma_{Cl-(aq)}$ are calculated in E-AIM, while only the mean binary activity coefficient of $\gamma_{H\text{-}Cl} = \sqrt{\gamma_{H+(aq)}\gamma_{Cl-(aq)}}$ is estimated in ISORROPIA.

**Global models.** Spatial variation of $c_{ni}$ was studied based on the two global models. The global GEOS-Chem model simulations (v11-01) were conducted at a resolution of 2.5° longitude × 2° latitude with 47 vertical layers for 2016. Detailed

model settings are provided elsewhere (Zheng et al., 2020). The global EMAC (ECHAM5/MESSy2 for Atmospheric Chemistry) model were conducted at a resolution of T63 (i.e., ~1.8° × 1.8° at the equator) with 31 vertical levels for 2016. Detailed EMAC model settings are provided in Appendix C.

**3. Results and discussion**

**3.1 Influencing factors of the non-ideality coefficient**

All activity coefficients first depends on RH and temperature. In addition, for ammonium-buffered ambient aerosols, major anions in pair with NH$_4^+$ or H$^+$ is NO$_3^-$ and SO$_4^{2-}$. The ratio of mean activity coefficients is therefore expected to differ when they're mainly combined with SO$_4^{2-}$ (i.e., $\gamma_{NH4HSO4}/\gamma_{H\text{-}HSO4}$) or NO$_3^-$ (i.e., $\gamma_{NH4NO3}/\gamma_{HNO3}$).

Figure 1 shows the dependence of $c_{ni}$ under different systems (Appendix A), as predicted by the gas-particle portioning (Eq. 14b) with E-AIM (Fig. 1 a, c, e) and ISORROPIA II (Fig. 1b, d, f), respectively. Based on both models, $c_{ni}$ differs much

between NH$_3$-H$_2$SO$_4$ system (Fig. 1a, b) and NH$_3$-HNO$_3$-H$_2$SO$_4$ system (Fig. 1c,d), even at the same RH and temperature. The difference is still large when compared at the same ionic strength and temperature (Fig. A1), illustrating that the difference is mainly due to the ion-pair specific binary activity coefficients, $\gamma_{ij}^o$ (Zaveri et al., 2005; Fountoukis and Nenes, 2007; Clegg et al., 1992b) (Appendix B; Fig. B1).

Due to the large difference in $c_{ni}$ between NH$_4$NO$_3$ and (NH$_4$)$_2$SO$_4$ dominated aerosols, the $c_{ni}$ at given RH and temperature

conditions is therefore sensitive to the anion profiles, as characterized by the fraction of NO$_3^-$ in anions(aq), $f_{NO3-}$, of:

$$f_{NO3-} \ (\mu eq/\mu eq) = [NO_3^-(aq)]/[Anions(aq)] \tag{15a}$$

$$[Anions(aq)] = 2 \ [SO_4^{2-}(aq)] + [NO_3^-(aq)] + [Cl^-(aq)] \tag{15b}$$

The $f_{NO3-}$ is proportional to NO$_3^-$/SO$_4^{2-}$ molar ratios when Cl$^-$ is negligible. In comparison, the cation profiles, or the relative abundances of non-volatile cations (NVCs; total cations from Na$^+$, Ca$^{2+}$, K$^+$, and Mg$^{2+}$), play a minor role as their influence

is more indirect (Fig. 1e, f).





## 3.2 Comparison of $c_{n,i}$ estimated by E-AIM and ISORROPIA

As discussed in section 2.2, for E-AIM $c_{ni}$ can be estimated either by activity coefficients (Eq. 14a) or gas-particle portioning (Eq. 14b), and the results agreed perfectly (black lines in Fig. 2). Therefore, the $c_{ni}$ estimation with E-AIM is calculated by the gas-particle portioning (Eq. 14b) hereinafter, the same as ISORROPIA.

Although showing the same influencing factors, $c_{ni}$ estimated by E-AIM and ISORROPIA are not identical (Fig. 1). Especially for the $NH_3$-$H_2SO_4$-$H_2O$ system (i.e., $(NH_4)_2SO_4$ dominated aerosols), E-AIM (Fig. 1a) and ISORROPIA (Fig. 1b) even predicted reversed trends in $c_{ni}$ dependence on RH and temperature. This is more clearly shown in Fig. 2 (blue dots), where $c_{ni}$ by E-AIM and ISORROPIA at the same conditions (i.e., same RH, temperature and chemical profiles) are compared. As shown in Fig.2a, while $c_{ni}$ predicted by E-AIM ranged -0.3 to 0.5 for $(NH_4)_2SO_4$ dominated aerosols, that by ISORROPIA is always

larger than 0.1.

For the $NH_3$-$H_2SO_4$-$H_2O$ system, we found that these two models generate similar prediction of AWC (and therefore similar ideal constant, $K_{a,NH3}^{*,i}$) (Fig. D1a). The different $c_{ni}$ is mainly due to disagreement in the predicted molar ratios of $NH_3(g)/NH_4^+$ (Fig. D1b). This is caused by the difference of calculated activity coefficients between ISORROPIA (Song et al., 2018; Fountoukis and Nenes, 2007) and E-AIM (see details in Appendix D and Fig. D2). Despite the difference in estimated $c_{ni}$, the

difference in pH predictions by E-AIM and ISORROPIA is relatively small, as pH was mainly controlled by $pK_{a,NH3}^{*,i}$ (Fig. D1c).

Unlike the $NH_3$-$H_2SO_4$-$H_2O$ system, $c_{ni}$ estimated by ISORROPIA generally agrees well with (while tends to be somewhat higher than) E-AIM when $HNO_3$ is present in the system (Fig. 2b, c). This indicates that constraint from $NH_3$-$HNO_3$ equilibriums are quite important in estimating $c_{ni}$ with ISORROPIA (see details in Appendix D). Under ambient conditions,

there's barely places with negligible $HNO_3$, thus the ISORROPIA predicted $c_{ni}$ generally agreed with E-AIM (section 3.4). With the known governing factors, here we propose a parameterization method to estimate $c_{ni}$ at given RH, temperature and $f_{NO3^-}$, with lookup tables generated by comprehensive thermodynamic models, E-AIM and ISORROPIA ("AIM_molality" database and "ISORROPIA_molality" database as in Data S1). In addition, the parameterized $c_{ni}$ for activity-based pH (Eq. 11; Appendix B; Fig. B1) is also avaialble ("AIM_activity" database in Data S1). A Matlab code to get $c_{ni}$ is also provided

(Data S1). Example slices of this $c_{ni}$ parameterization based on "AIM_molality" estimations are shown in Fig. 3. Note that this parameterization method aimed at $NH_3$-$HNO_3$-$H_2SO_4$-$H_2O$ system, assuming no NVCs. We will show that this assumption is acceptable under most cases in the following sections.

## 3.3 Validation and applications with long-term observations

To validate the $c_{ni}$ parameterization method under actual ambient conditions, we here show an example application based on

the long-term measurements in Toronto (Tao and Murphy, 2019) (Fig. 4). From 2007 to 2016, Toronto resides in the ammonia-buffered regime for ~80% of the times, and the model-predicted pH based on the measured chemical compositions follows nicely with the variation of actual $pK_a^{*, ni}$ estimated by thermodynamic models (Eq. 14), for both E-AIM (Fig. 4a) and



ISORROPIA (Fig. 4c). Parameterized $c_{ni}$ agreed quite well with the actual ones for both models (Fig. 4b, d, black circles), with $R^2$ both being 0.99, and the corresponding root-mean-square deviation (RMSD) both being ~0.1.

Figure 4 also suggest that most of the variation of actual $pK_a^{*,ni}$ comes from the variation of ideal constants ($pK_a^{*,i}$), not the non-ideality. For example, assuming the full aerosol and gas measurements were conducted only in a calibration year of 2012, based on which the annual mean and monthly mean $c_{ni}$ can be derived (Fig. E1). Annual mean $c_{ni}$ is 0.4 for E-AIM and 0.8 for ISORROPIA estimations. When we use the annual mean $c_{ni}$ as a constant correction (i.e., estimated $pK_a^{*,ni}$ = $pK_a^{*,i}$ + annual mean), fluctuation in the estimated $pK_a^{*,ni}$ would actually all come from $pK_a^{*,i}$. However, this estimated $pK_a^{*,ni}$ can already

explain ~90 % of the variations in actual $pK_a^{*,ni}$ (red dots in Fig. 4b, d), illustrating the dominance of $pK_a^{*,i}$ (i.e., AWC and temperature fluctuations) over non-ideality. In comparison, applying the month-dependent $c_{ni}$ values (blue dots in Fig. 4b, d) makes little difference with the annual constant estimations ($R^2$ differed only by 1%).

Figure 4 and Fig. E1 illustrate that a constant $c_{ni}$ is often good enough at a given site. Full aerosol species measurements for a whole year, or under periods representative of annual-average conditions (like spring or fall seasons for Toronto; Fig. E1) is

recommended in determining the localized $c_{ni}$, which, together with AWC and temperature measurements, could already provide a good approximation of the aerosol pH. This is especially useful in retrieving the acidity variations when full chemical measurements are not available in the long run.

### 3.4 Validation and application against global model simulations

We further investigated the influence of non-ideality in explaining the spatial variations of aerosol acidity based on global

model simulations. On the global scale, fraction of $NO_3^-$ in aqueous phase anions depends on two factors: the total nitrate (gas + particle phase) to sulfate ratios, and the partitioning of total nitrates. When total nitrate << sulfates, the aerosols would be dominated by $(NH_4)_2SO_4$ even if all the nitrates are partitioned into the particle phase. In this case, non-ideality correction factor can be estimated from Fig. 1a, b at known RH and temperature. However, both GEOS-Chem and the EMAC results show that this criterion is barely met for the ammonia-buffered regions. Besides, for all the reported observation results we

know of, only summertime south-eastern U.S. (Weber et al., 2016) has a total nitrate that is < 5% of the sulfate (charge ratios). Therefore, under most conditions, $c_{ni}$ largely depends on the partitioning of total nitrates, and an estimation of $f_{NO3-}$ is needed to derive the correction factor.

Figure 5 shows the estimated $pK_a^{*,ni}$ against actual $pK_a^{*,ni}$ based on GEOS-Chem simulations, and that based on EMAC simulations are shown in Fig. C1. Three scenarios are assumed to examine the sensitivity of $pK_a^{*,ni}$ prediction with $c_{ni}$ values:

(a) constant temperature ($T$) of 288 K and RH of 73%, (b) constant RH of 73%, but with annual-average temperatures for each site; and (c) annual-average $T$ and RH for each site. For all scenarios, annual mean chemical compositions for the ammonia-buffered surface regions (Zheng et al., 2020) are used, and $c_{ni}$ is estimated by both E-AIM and ISORROPIA II models. Similar with Fig. 4, in the "parameterized" series $c_{ni}$ is estimated by the parameterization method proposed in this study with RH, $T$ and $f_{NO3-}$ at certain model grid, while in the "global mean" series, $c_{ni}$ is assumed to be constant as the average of actual $c_{ni}$





estimated by the thermodynamic models under each scenario, which is ~0.6 for E-AIM model and ~0.8 for ISORROPIA model.

Based on GEOS-Chem simulations, the parameterized $c_{ni}$ (black dots in Fig. 5) work nicely in reproducing actual $pK_a^{*,ni}$, with $R^2$ near 1 under all scenarios, and the RMSD of <0.03 for ISORROPIA model and ~0.1 for AIM model. Again, we found that variations of $c_{ni}$ is much smaller than the variation of $pK_a^{*,i}$ caused by particulate matter concentrations and temperatures. With

a constant global-mean $c_{ni}$ correction (i.e., assuming a global average $f_{NO3-}$) (blue dots in Fig. 5), the estimated $pK_a^{*,ni}$ can already explain over 93 % of the variations in actual $pK_a^{*,ni}$, with/without considering the influence of meteorology on non-ideality alike. Correspondingly, it can already explain ~70% of the aerosol pH variations (Zheng et al., 2020), where the pH is further subject to variations in $NH_3(g)$ and NVCs (Eq. 8; Zheng et al. (2020)).

The EMAC simulations show similar patterns with GEOS-Chem results. Estimated $pK_a^{*,ni}$ with the parameterized $c_{ni}$

corrections agreed well with actual $pK_a^{*,ni}$, with $R^2$ over 0.94 for E-AIM model and over 0.91 for ISORROPIA model (Fig. C1). This is somewhat lower than the Toronto site (Fig. 4) or the GEOS-Chem result (Fig. 5), which is due to the larger variations in the simulated chemical profiles (e.g., importance of NVCs and Cl⁻, etc.). The constant $c_{ni}$ assumption (blue dots in Fig. C1) works similarly with the parameterized ones when influence of meteorology is excluded (Fig. C1 a, d) or when spatial variations of temperatures are considered (Fig. C1 b,e). When spatial variations of both temperature and RH (Fig. C1c,f)

are considered, the constant $c_{ni}$ assumption works worse than the parameterized ones, but is still acceptable ($R^2$ being 0.75 for E-AIM and 0.69 for ISORROPIA).

Note that under all conditions, the "global mean" method tend to overestimate $c_{ni}$ when actual $pK_a^{*,ni}$ of $NH_3$ is smaller than 2 (Fig. 5, Fig. C1). That is caused by $f_{NO3-}$. The low $pK_a^{*,ni}$ indicates low AWC levels (Zheng et al., 2020) and relatively low pH levels (Eq. 13). Under such conditions, $HNO_3$ tends to stay in the gas phase (Nenes et al., 2020), corresponding to a low $f_{NO3-}$

of ~0. In comparison, the global-mean $c_{ni}$ correspond to the global-mean simulated $f_{NO3-}$ of ~0.4. As $c_{ni}$ increase with increasing $f_{NO3-}$ (Fig. 1e,f), the global-mean $c_{ni}$ would tend to overestimate actual low $pK_a^{*,ni}$ conditions (i.e., < ~2).

## 4. Conclusions

Overall, we found that the non-ideality correction is needed for using $pK_a^{*,ni}$ of $NH_3$ as a proxy of aerosol pH in ammonia-buffered regimes. This correction factor, $c_{ni}$, generally ranging 0.3 ~ 1.1, and mainly depends on RH, temperature and the

fraction of nitration in aqueous-phase anions. E-AIM generally predicted a lower $c_{ni}$ than the ISORRPIA model. We proposed a parameterization method to estimate the $c_{ni}$, which works quite well, as validated against both long-term observations and global simulations. Although the correction is needed in estimating the ammonia $pK_a^{*,ni}$ levels, the variations in $pK_a^{*,ni}$ is often much less sensitive to the non-ideality than to aerosol water content and temperature. Therefore, a constant correction factor of $pK_{a,NH_3}^*$ is often good enough to predict pH over a period at a given site, or to explain the global pH variations. We thereby

provided a way for pH retrieval when chemical measurements are unavailable for the ammonia-buffered regions.



*Data availability.* All data used in this study are described in the manuscript and supporting information.

**Appendix A. Scenario settings for different systems**

In Figs. 1, 2, A1, B1, D1and D2, we assumed three systems, with the settings detailed below.

**NH₃-H₂SO₄-H₂O system.** For this system, we assumed a constant input with 0.5 $\mu$mol m$^{-3}$ of total sulfate (i.e., 1 $\mu$eq m$^{-3}$ of anions) and 2 $\mu$mol m$^{-3}$ of total ammonia. This ratios is to ensure that the system pH is around the maximum buffering capacity of ammonia. However, we found that for ISORROPIA model, the solver with only ammonia and sulfates inputs is not stable,

with predicted pH often larger than 7 (SI S4). We thereby introduced 0.015 $\mu$mol m$^{-3}$ of Na$^+$ (3% of the total sulfate molar concentrations, or 1.5% of the anions), which exerted little influence on the ionic environments (difference in E-AIM results less than 3%) but will change the ISORROPIA subroutine solver called. The RH and temperature are then varied at different values to check the influence.

**NH₃-HNO₃-H₂O system.** For this system, we assumed a constant input with 1 $\mu$mol m$^{-3}$ of total nitrate (also 1 $\mu$eq m$^{-3}$ of

anions) and 2 $\mu$mol m$^{-3}$ of total ammonia, and then varied the RH and temperatures to derive non-ideality correction factors.

**Na$^+$-NH₃-HNO₃-H₂SO₄-H₂O system.** For this system, we fixed the RH at 73% and temperature at 288.15 K, 2 $\mu$mol m$^{-3}$ of total ammonia and a fixed concentration of total anions as 1 $\mu$eq m$^{-3}$. The nitratie/sulfate ratios are then varied (but keeping their total charges the same) to get different nitrate fractions. For example, when the input sulfate is 0.25 $\mu$mol m$^{-3}$ equalling 0.5 $\mu$eq m$^{-3}$ of anions, the input total nitrate is then set to 0.5 $\mu$mol m$^{-3}$, corresponding to a total anion of the system as 1 $\mu$eq

m$^{-3}$. Meanwhile, the ratio of NVCs (here assumed to be Na$^+$ only) to anions is also varied, and combined with the different nitrate/sulfate ratios to generate different simulation conditions.

**Appendix B: Non-ideality correction factor for activity-based pH definitions**

With activity-based pH definition (i.e., pH=- log ($\gamma_{H+}$ [H$^+$]), multiphase buffer theory can be rewritten as:

$$K_{a,NH3}^{*,nia} = \frac{[NH_3(g)]}{[NH_4^+(aq)]}(\gamma_{H+}[H^+(aq)]) = \gamma_{NH4+}K_{a,NH3}^{*,i} = \gamma_{NH4+}K_{a,NH3}(1+\frac{\rho_w}{H_{NH3} \, R \, T \, AWC}) \tag{A1a}$$

$$pH = -\log(\gamma_{H+}[H^+(aq)]) = pK_{a,NH3}^{*,nia} + \log\frac{[NH_3(g)]}{[NH_4^+(aq)]} \tag{A1b}$$

where $K_a^{*,nia}$ is the multiphase effective acid dissociation constant under non-ideal conditions. The difference of pH caused by non-ideality, $c_{nia}$, is therefore:



$$c_{nia} = \mathrm{p}K_a^{*,nia} - \mathrm{p}K_a^{*,i} = -\log \gamma_{NH4+} \tag{A2}$$

That is, the non-ideality correction factor for activity-based pH is actually the $\gamma_{NH4+}$, which can be calculated with the more comprehensive models like E-AIM. The E-AIM calculated mole-fraction-based activity coefficient ($f_i$) that can be converted to the molality-based activity coefficient ($\gamma_i$) by (Pye et al., 2020; Peng et al., 2019):

$$\gamma_i = f_i x_w = f_i x_i / (m_i M_w) \tag{A3}$$

where $x_i$ and $m_i$ are respectively the mole fraction and molality of species $i$, and $x_w$ and $M_w$ are respectively the mole fraction and molecular weight of water. All these variables are given in E-AIM outputs. Major influencing factors of $c_{nia}$ is also RH, temperature and fraction of $NO_3^-$ in anions in the aqueous phase (aq), as shown in Fig. B1.

**Appendix C. EMAC model settings**

In this section, we will only focus on the model settings for EMAC simulations, while for the GEOS-chem model settings, please refer to Zheng et al. (2020). We used the global ECHAM/MESSy Atmospheric chemistry – Climate (EMAC) model, which is a numerical chemistry and climate simulation system that includes sub-models describing tropospheric and middle atmosphere processes and their interaction with oceans, land and human influences (Jöckel et al., 2010). The core atmospheric model is the 5th generation European Centre Hamburg general circulation model (ECHAM5) (Roeckner et al., 2006), which has been modularized, and to which improved submodels and updates of boundary layer, radiation, cloud and convection routines have been introduced. The EMAC model development is coordinated within an international consortium: see https://www.messy-interface.org. For the present study we applied EMAC (ECHAM5 version 5.3.02, MESSy version 2.54.0) in the T63L31-resolution, i.e., with a spherical truncation of T63 (corresponding to a quadratic Gaussian grid of approx. 1.8 by 1.8 degrees in latitude and longitude) with 31 vertical hybrid terrain-following pressure levels up to 10 hPa in the lower stratosphere. Meteorological conditions as in ERA-interim data from European Centre for Medium- range Weather Forecasts (ECMWF) were simulated by the model by applying a "nudging" technique (Jöckel et al., 2006). EMAC simulates gas-phase and heterogeneous chemistry through the MECCA submodel, which accounts for the photochemical oxidation of natural and anthropogenic emissions, including a comprehensive account of volatile organic carbon compounds (Sander et al., 2019). Aerosol microphysical processes and gas/particle partitioning are simulated with the GMXe submodel (Pringle et al., 2010; Pozzer et al., 2012), which describes the aerosol size distribution by seven interacting lognormal modes (four hydrophilic and three hydrophobic). The aerosol composition can vary between these modes (externally mixed) and is uniform within each mode (internally mixed). The inorganic aerosol composition is computed with the ISORROPIA-II thermodynamic equilibrium submodel (Fountoukis and Nenes, 2007). It calculates the gas/liquid/solid equilibrium partitioning of inorganic compounds and water. The composition and atmospheric evolution of organic aerosol compounds are simulated with the ORACLE submodel, which represents volatility classes of organics through their effective saturation concentrations (Tsimpidi et al., 2018). For this work the anthropogenic emissions EDGAR (Emissions Database for Global Atmospheric Research v4.3.2)



(Crippa et al., 2018) were applied, as well as the GFAS (Global Fire Assimilation System, v1.0) (Kaiser et al., 2012) for
biomass burning emissions. The EMAC results are shown in Fig. C1.

## Appendix D. Potential reasons for discrepancies in predicting aerosol pH by ISORROPIA and E-AIM for the $NH_3$-$H_2SO_4$-$H_2O$ system

In this study, we applied ISORROPIA version 2.1 (Fountoukis and Nenes, 2007) and E-AIM (model IV; http://www.aim.env.uea.ac.uk/aim/aim.php) (Clegg et al., 1992b; Wexler and Clegg, 2002; Friese and Ebel, 2010), and the
following description and discussion refer to these versions of the models. For $NH_3$-$H_2SO_4$-$H_2O$ system, we found that by assuming the same input of 0.5 $\mu$mol m$^{-3}$ of total sulfate (i.e., 1 $\mu$eq m$^{-3}$ of anions) and 2 $\mu$mol m$^{-3}$ of total ammonia and varies the RH (60% - 90%) and temperatures (265 K - 300 K), ISORROPIA predicted a very high aerosol pH of about 13 (12.6-13.2) while the E-AIM predicted pH ranged 2~5, which is obviously more realistic. However, by introducing only an small amount of Na$^+$ (0.015 $\mu$mol m$^{-3}$, or 3% of the total sulfate), the ISORROPIA predicted pH dropped dramatically to 2~5 (Fig. D1),
while the E-AIM predicted pH changed little than the no-Na$^+$ predictions (pH increased systematically by 0.03 with both R$^2$ and slope being 1). Besides, the predicted pH assuming only HNO$_3$ and NH$_3$ inputs (NH$_3$-HNO$_3$-H$_2$O system) agreed well between ISORROPIA and E-AIM (Fig. D1).

We found that the dramatic changes in ISORROPIA predicted pH levels with or without small amount of Na$^+$ and NO$_3^-$ additions are related to the different calculation procedures among subcases. Here we focused on subcases under the metastable
and sulphate-poor (i.e., total potential cations, including total ammonia ([NH$_3$]$_t$) and NVCs, exceed twice the molar ratios of total sulfate ([H$_2$SO$_4$]$_t$)) conditions.

In ISORROPIA, when there's only NH$_3$ and H$_2$SO$_4$ (i.e., "pure" NH$_3$-H$_2$SO$_4$-H$_2$O system), the corresponding subcase is "A2". As detailed below, for this subcase, activity coefficients included in the final calculations are $\gamma_{H-HSO4}$, $\gamma_{2H-SO4}$, and $\gamma_{NH4-HSO4}$. As shown in Fig. D2a-c, for all these three values, there's large difference between E-AIM and ISORROPIA estimations (note
that log scales are used for $\gamma_{H-HSO4}$ and $\gamma_{2H-SO4}$ plots). Therefore, it's not surprising that there is large discrepancy between the predicted pH from subcase A2 of ISORROPIA and E-AIM.

In comparison, the subcase would change to "D3" when HNO$_3$ is introduced to the system. As detailed below, for this subcase, only $\gamma_{NH4-NO3}$ is involved in the calculations. As shown in Fig. D2d, although the ISORROPIA still shows a different trend than the E-AIM, it is, however, at least on the same order of magnitude as the one predicted by E-AIM.

By introducing a small amount of Na$^+$ into the NH$_3$-H$_2$SO$_4$-H$_2$O system, the calculation procedure of ISORROPIA would change from A2 to G5 (a Na$^+$-NH$_3$-H$_2$SO$_4$-HNO$_3$-HCl-H$_2$O aerosol system). For G5 subcase, we noticed two issues: (1) although the total HNO$_3$ is zero, the model still tried to predict $\gamma_{H-NO3}$ and $\gamma_{NH4-NO3}$; (2) as it was using Cl$^-$ as the $x$ variable at the final solutions, a small amount of Cl$^-$ is always present, which is introduced by the model so the calculation procedures could go on. The relevant values are shown in Fig. D2e. In comparison, the E-AIM predicted no NO$_3^-$ or Cl$^-$, and the activity
coefficients of other relevant species change little with the no Na$^+$ case. Therefore, we could not perform a comparison between





ISORROPIA and E-AIM for this case (as there's no $\gamma_{NO3}$ or $\gamma_{Cl}$ in E-AIM). Based on the pH and non-ideality comparisons (Fig. D1), however, we could see that the $NH_3$ partitioning estimated this way is far more realistic than the A2 subcase.

**Calculation principles for subcase A2 (an NH₃-H₂SO₄-H₂O aerosol system).** For the subcase A2, the major constraining
equations include the $[SO_4^{2-}]/[HSO_4^-]$ equilibriums, gas-particle partitioning of ammonia, and charge balance:

$$HSO_4^- \rightleftharpoons H^+ + SO_4^{2-}, \quad K_{a,HSO4} = \frac{[H^+(aq)][SO_4^{2-}(aq)]}{[HSO_4^-(aq)]} \frac{\gamma_{H+(aq)}\gamma_{SO42-(aq)}}{\gamma_{HSO4-(aq)}} = \frac{[H^+(aq)][SO_4^{2-}(aq)]}{[HSO_4^-(aq)]} \frac{\gamma^3_{2H-SO4(aq)}}{\gamma^2_{H-HSO4(aq)}} \tag{D1a}$$

$$NH_4^+ \rightleftharpoons NH_3(aq) + H^+, \quad K_{ag,NH3} = K_w / K_{bg,NH3} = \frac{[H^+(aq)][NH_3(g)]\gamma_{H+(aq)}}{[NH_4^+(aq)]\gamma_{NH4+(aq)}} = \frac{[H^+(aq)][NH_3(g)]}{[NH_4^+(aq)]} \frac{\gamma^2_{H-HSO4(aq)}}{\gamma^2_{NH4-HSO4(aq)}} \tag{D1b}$$

$$([NH_4^+] + [H^+]) / (2[SO_4^{2-}] - [HSO_4^-]) - 1 = 0 \tag{D1c}$$

With these three equations and known total ammonia ($[NH_3]_t$) and total sulfate ($[H_2SO_4]_t$), we have:

$$[NH_3]_t\, C_{2S}[H^+] / (1+ C_{2S}[H^+]) + [H^+] - [H_2SO_4]_t (2\, C_1 / [H^+] + 1) / (1 + C_1 / [H^+]) = 0 \tag{D2}$$

where $C_1 = K_{a,HSO4} \dfrac{\gamma^2_{H-HSO4(aq)}}{\gamma^3_{2H-SO4(aq)}}$ while $C_{2S} = \dfrac{\gamma^2_{H-HSO4(aq)}}{K_{ag,NH3}\gamma^2_{NH4-HSO4(aq)}}$. The only unknown is thus $[H^+]$, which can thus be solved by

bisection solution processes. As shown in the equation, activity coefficients that matters in solving this system include

$\dfrac{\gamma^2_{H-HSO4(aq)}}{\gamma^3_{2H-SO4(aq)}}$ in $C_1$ and $\dfrac{\gamma^2_{H-HSO4(aq)}}{\gamma^2_{NH4-HSO4(aq)}}$ in $C_{2S}$.

**Calculation principles for subcase D3 (an NH₃-H₂SO₄-HNO₃-H₂O aerosol system).** For the subcase D3, the major
equilibriums considered is the gas-particle partitioning of ammonia and nitrates of:

$$NH_4^+ \rightleftharpoons NH_3(aq) + H^+, \quad K_{ag,NH3} = K_w / K_{bg,NH3} = \frac{[H^+(aq)][NH_3(g)]\gamma_{H+(aq)}}{[NH_4^+(aq)]\gamma_{NH4+(aq)}} = \frac{[H^+(aq)][NH_3(g)]}{[NH_4^+(aq)]} \frac{\gamma^2_{H-NO3(aq)}}{\gamma^2_{NH4-NO3(aq)}} \tag{D3}$$

$$HNO_3(g) \rightleftharpoons H^+ + NO_3^-, \quad K_{ag,HNO3} = \frac{[H^+(aq)][NO_3^-(aq)]\gamma_{H+(aq)}\gamma_{NO3-(aq)}}{[HNO_3(g)]\gamma_{HNO3(g)}} \tag{D4}$$

Note that in subcase D3 the $\gamma_{H+}/\gamma_{NH4+}$ is estimated by $(\gamma_{H-NO3}/\gamma_{NH4-NO3})^2$, not the $(\gamma_{H-HSO4}/\gamma_{NH4-HSO4})^2$ as in subcase A2.
These two equilibriums are further combined to be:

$$C_3 = \frac{[NO_3^-(aq)][NH_4^+(aq)]}{[HNO_3(g)][NH_3(g)]} = \frac{K_{ag,HNO3}}{K_{ag,NH3}\gamma^2_{NH4-NO3}} \tag{D5a}$$

As to the charge balance, here only major species are considered as:

$$[NH_4^+(aq)] = 2[SO_4^{2-}(aq)] + [NO_3^-(aq)] = 2[H_2SO_4]_t + [NO_3^-(aq)] \tag{D5b}$$





Combining Eqs. D5a and D5b, at given total nitrate ([HNO$_3$]$_t$, namely [NO$_3^-$(aq)] + [HNO$_3$(g)]) and [NH$_3$]$_t$ (= [NH$_4^+$(aq)] + [NH$_3$(g)]) levels, the solution function can be expressed as:

$$\frac{[NH_4^+(aq)]}{[NH_3(g)]}\frac{[NO_3^-(aq)]}{[HNO_3(g)]} - C_3 = 0$$

$$\frac{[NH_4^+(aq)]}{([NH_3]_t - [NH_4^+(aq)])}\frac{([NH_4^+(aq)] - 2[H_2SO_4]_t)}{([HNO_3]_t + 2[H_2SO_4]_t - [NH_4^+(aq)])} - C_3 = 0 \tag{D6}$$

Where the only unknown is [NH$_4^+$(aq)] and can be solved through bisection method. As shown in the equation, the only activity coefficients that matters in solving this system is $(\gamma_{NH4\text{-}NO3})^2$ in $C_3$.

**Calculation principles for subcase G5 (a Na$^+$-NH$_3$-H$_2$SO$_4$-HNO$_3$-HCl-H$_2$O aerosol system).** For the subcase G5, the major

equilibriums considered is the gas-particle partitioning of NH$_3$, HNO$_3$ and HCl, while sulfate is considered to exist mainly as [SO$_4^{2-}$(aq)]. General derivation processes are similar with D3 and is also detailed in a previous study (Song et al., 2018). Briefly, the key equilibriums include that of HNO$_3$ (Eq. D4) and HCl of:

$$HCl(g) \rightleftharpoons H^+(aq) + Cl^-(aq), \quad K_{ag,HCl} = \frac{[H^+(aq)][Cl^-(aq)]\gamma_{H+(aq)}\gamma_{Cl-(aq)}}{[HCl(g)]\gamma_{HCl(g)}} = \frac{[H^+(aq)][Cl^-(aq)]\gamma^2_{H\text{-}Cl}}{[HCl(g)]} \tag{D7}$$

Which can be combined into:

$$C_4 = \xi_{HNO3} / \xi_{HCl} = \frac{[NO_3^-(aq)][HCl(g)]}{[Cl^-(aq)][HNO_3(g)]} = \frac{K_{ag,HNO3}}{K_{ag,HCl}}\frac{\gamma^2_{H\text{-}Cl}}{\gamma^2_{H\text{-}NO3}} \tag{D8}$$

Therefore [NO$_3^-$(aq)] and [HNO$_3$(g)] (=[HNO$_3$]$_t$ - [NO$_3^-$(aq)]) can be solved at known assumed [Cl$^-$(aq)].

And the [NH$_4^+$(aq)] associated with Cl$^-$(aq) and NO$_3^-$(aq), [NH$_4^+$(aq)]$_{NC}$, is solved by:

$$([NH_4^+(aq)]_{NC})^2 - B\,[NH_4^+(aq)]_{NC} + C = 0 \tag{D9}$$

where

B = [NH$_3$]$_t$ + [Na$^+$] - 2[H$_2$SO$_4$]$_t$ + [Cl$^-$(aq)] + [NO$_3^-$(aq)] + $C_{2N}^{-1}$

C = ([NH$_3$]$_t$ + [Na$^+$] - 2[H$_2$SO$_4$]$_t$) ([Cl$^-$(aq)] + [NO$_3^-$(aq)]) - $C_{2N}^{-1}$ (2[H$_2$SO$_4$]$_t$ - [Na$^+$])

where $C_{2N} = \dfrac{\gamma^2_{H\text{-}NO3(aq)}}{K_{ag,NH3}\gamma^2_{NH4\text{-}NO3(aq)}}$ . And with [NH$_4^+$(aq)]$_{NC}$, we have:

[NH$_4^+$(aq)] = [NH$_4^+$(aq)]$_{NC}$ + 2[H$_2$SO$_4$]$_t$ - [Na$^+$]

[NH$_4^+$(g) = [NH$_3$]$_t$ - [NH$_4^+$(aq)]

The system then solves the equation sets through bisection method by assuming a series of [Cl$^-$(aq)] levels.

As shown in the equations above, activity coefficients that matters in solving this system (Eqs. D8~D9) include $\gamma_{H\text{-}NO3}$, $\gamma_{H\text{-}Cl}$, and $\gamma_{NH4\text{-}NO3}$.



**Appendix E. Information for the assumed calibration year of 2012 in Toronto site (Fig. E1).**


*Author contributions.* Y.C., H.S. and G.Z. designed the study. G.Z. performed the study. S.W. provided GEOS-Chem model results. A.P. provided the global numerical EMAC model results. G.Z., Y.C, and H.S. wrote the manuscript with input from all coauthors.

*Competing interests.* The authors declare no competing interests.

*Acknowledgments.* The research was supported by the Max Planck Society (MPG). Y.C. acknowledges the Minerva Program of MPG. We would like to thank Ulrich Pöschl for the helpful discussion and support.

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

**Figures**

**Figure 1. The non-ideality correction factor, $c_{ni}$, estimated by E-AIM (a, c, e) and ISORROPIA (b, d, f) for different aerosol systems.**
(**a, b**) NH₃-H₂SO₄-H₂O system with aerosols dominated by (NH₄)₂SO₄ at varying RH and temperature conditions; (**c, d**) NH₃-HNO₃-H₂O system with aerosols doeminated by NH₄NO₃ at varying RH and temperature conditions, and (**e, f**) Na⁺-NH₃-HNO₃-H₂SO₄-H₂O system with

varying chemical profiles at 288.15 K and RH of 73%. The chemical profiles in (e, f) are characterized by the fraction of NO₃⁻ in anions(aq) and NVCs/anions(aq), where the non-volatile cations (NVCs) are assumed to be Na⁺ only here. The assumed RH and T conditions in (e, f) are marked as blacked stars in (a-d), while the chemical profiles for (a-d) and (e, f) are marked by the corresponding letter in (e) and (f), respectively.

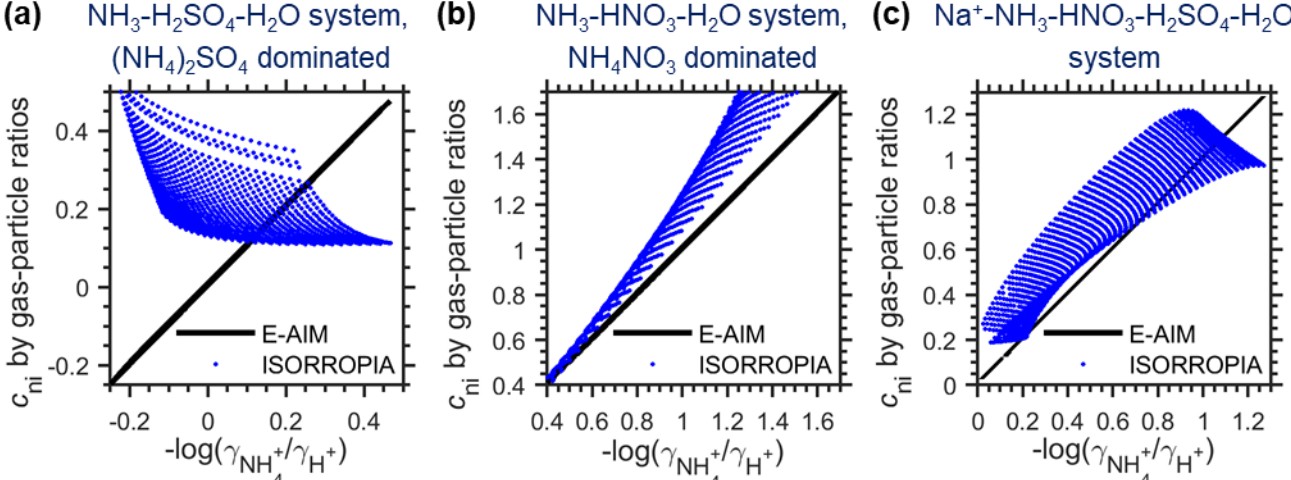

**Figure 2. Comparison of different $c_{ni}$ estimation methods for three representative aerosol systems.** The $c_{ni}$ are compared at the same conditions (i.e., same RH, temperature and chemical profiles). The $x$ values are $c_{ni}$ estimated by activity coefficients (Eq. 13a) with E-AIM model, and the $y$ values include $c_{ni}$ estimated by gas-particle ratios (Eq. 13b) with E-AIM (black lines) and ISORROPIA (blue dots) models. The systems are the same as Fig. 1.





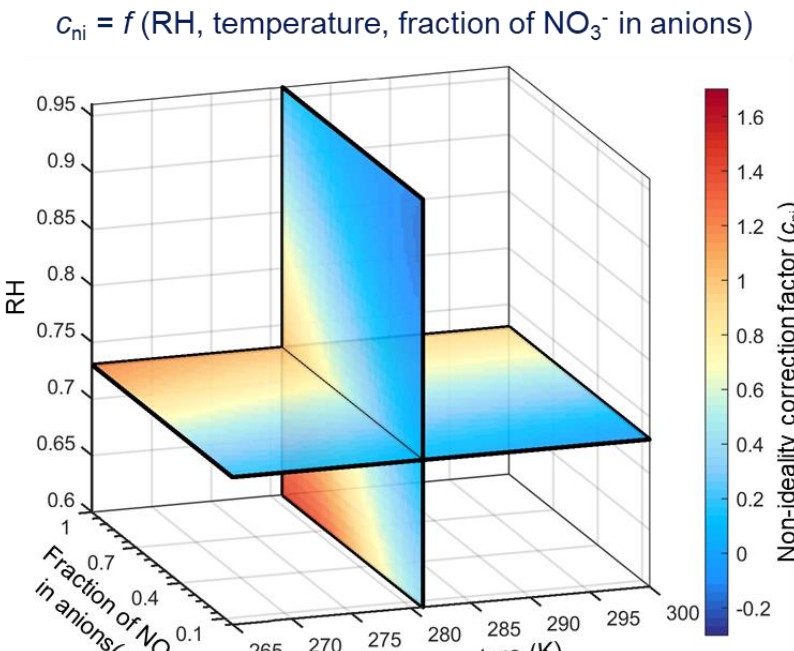


**Figure 3. Example slices of the $c_{ni}$ parameterization based on "AIM_molality" estimations as given in Data S1.**

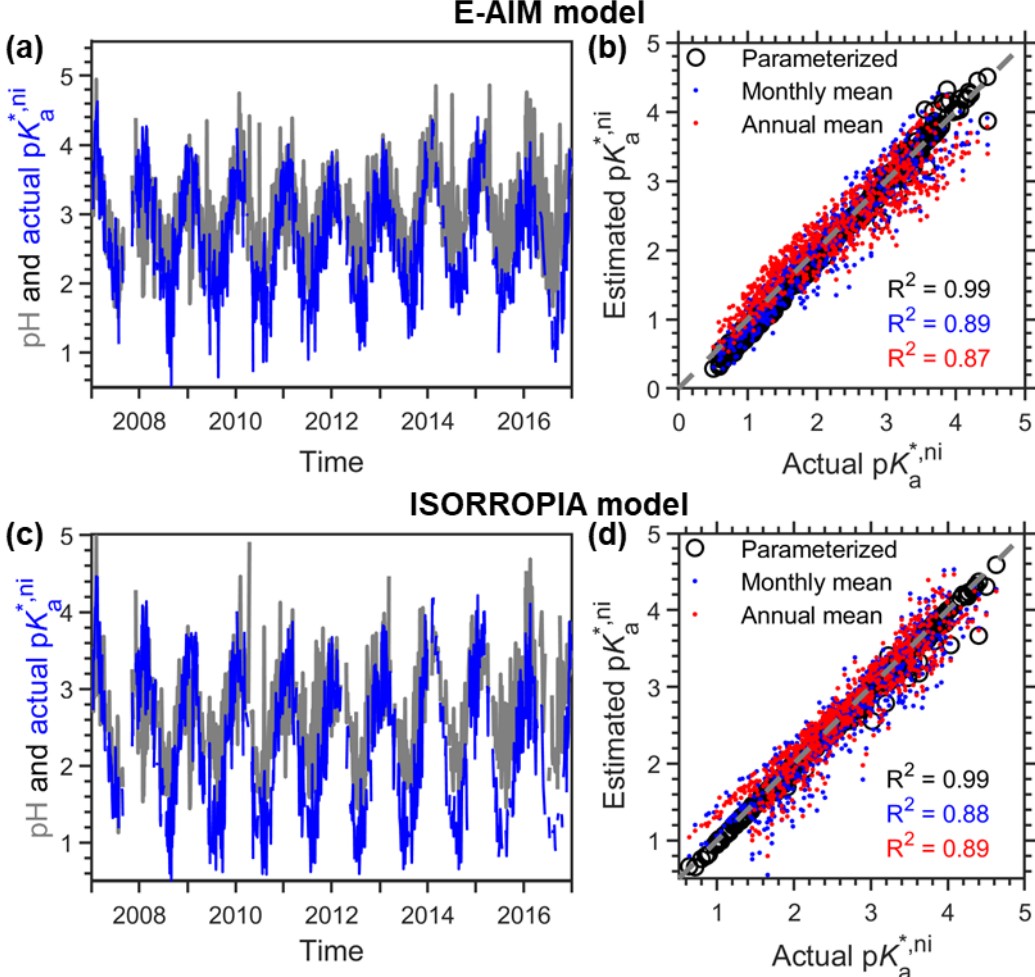

**Figure 4. Comparison of pH, actual and estimated p$K^*_{a,ni}$ based on the ten-year observations in Toronto.** Data were taken from
Canada's National Air Pollution Surveillance (NAPS) Program, as detailed in ref. [14]. Predications are based on (a-b) E-AIM model and (c-d) ISORROPIA model. The "parameterized" series in (b, d) are predicted by the parameterization method proposed with input of the observed RH, temperature and model predicted fraction of nitrates in anions. The annual mean and monthly mean are based on mean $c_{ni}$ of an arbitrary example year of 2012.



**Figure 5. Comparison of actual and estimated p$K^{*}_{a,ni}$ based on the GEOS-Chem global simulations in 2016.** Predications are based on (a-b) E-AIM model and (c-d) ISORROPIA model. The "parameterized" series are based on the parameterization method proposed in this study, while the global means are based on mean $c_{ni}$ calculated from thermodynamic models under each scenario.



**Figures for Appendices**

**Figure A1. Ionic strength (*I*) and the non-ideality correction factor, *c*ni, as calculated by E-AIM (a, c, e) and ISORROPIA (b, d, f) under different aerosol systems.** The systems are the same as Fig. 1, while the RH in Fig. 1(a-d) and *c*ni in Fig. 1(e-f) are replaced into *I*.





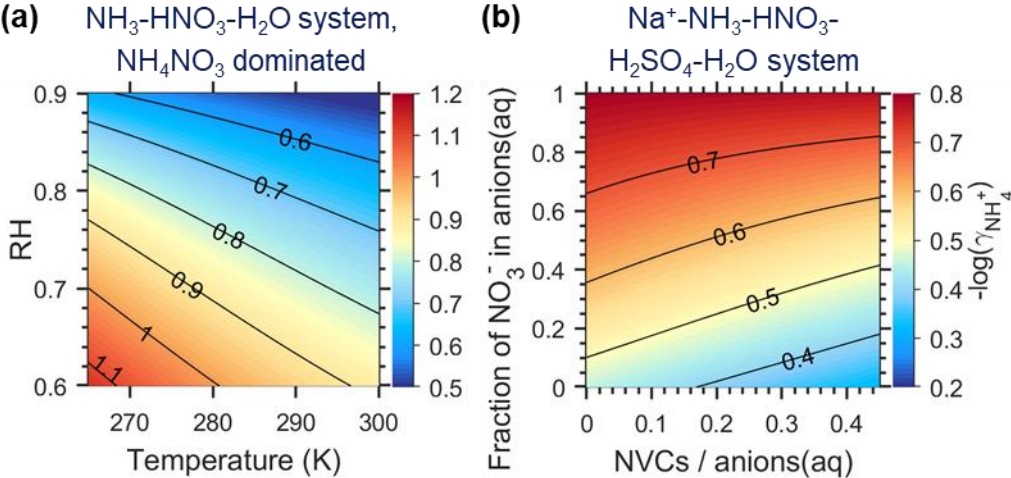


**Figure B1. Dependence of the non-ideality correction factor for activity-based pH definitions, $c_{nia}$ (i.e., -log($\gamma_{NH4}$)), as estimated by E-AIM.** (a) $NH_3$-$HNO_3$-$H_2O$ system with aerosols doeminated by $NH_4NO_3$ at varying RH and temperature conditions, and (b) $Na^+$-$NH_3$-$HNO_3$-$H_2SO_4$-$H_2O$ system with varying chemical profiles at 288.15 K and RH of 73%. Note that the $c_{nia}$ for $NH_3$-$H_2SO_4$-$H_2O$ system (i.e., $(NH_4)_2SO_4$-dominated aerosols) is not shown, as it varied little (ranging 0.44~0.47) over the whole RH and temperature ranges explored.


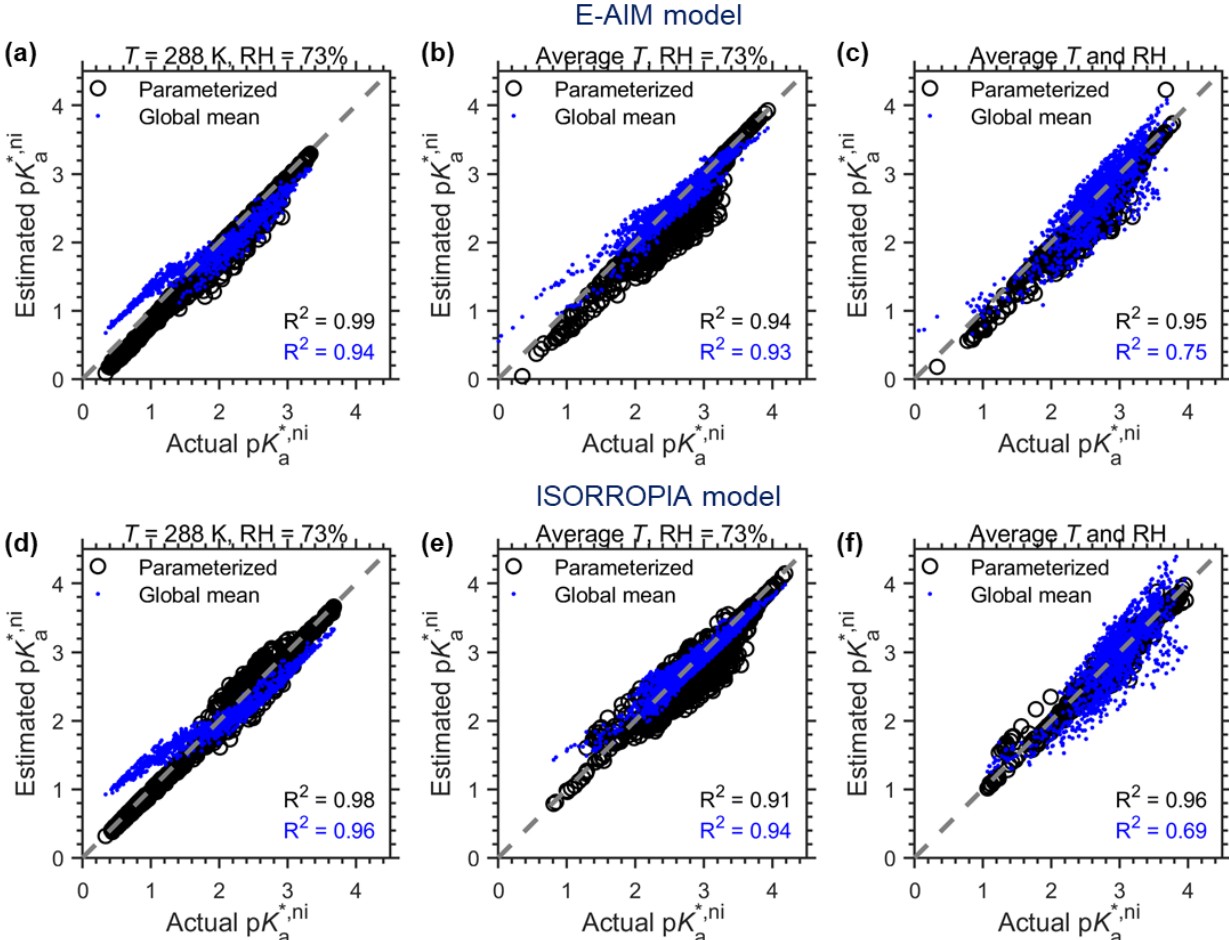

**Figure C1. Same as Fig. 5, but based on EMAC results.**



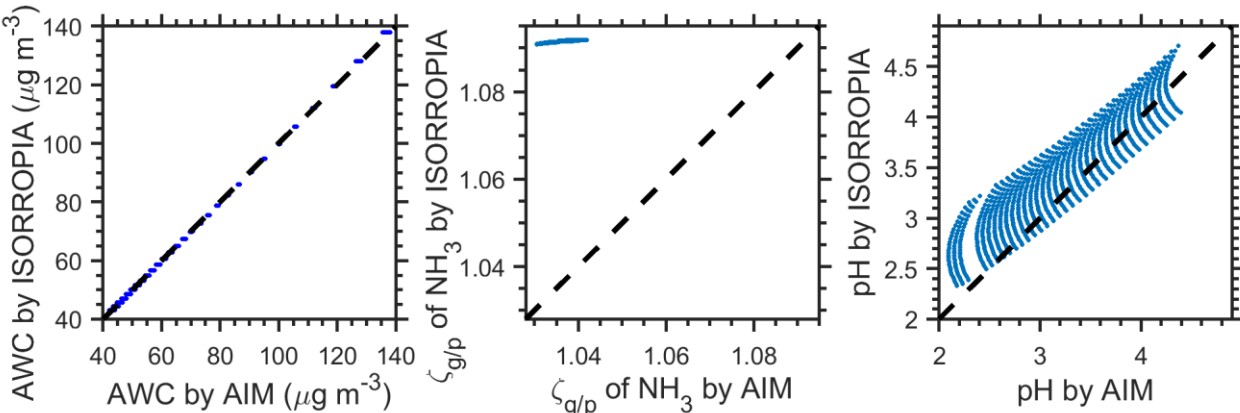

**Figure D1. Drivers of the difference in $c_{ni}$ estimated by ISORROPIA and E-AIM models for the NH$_3$-H$_2$SO$_4$-H$_2$O system.** The $\zeta_{g/p}$ of NH$_3$ indicates the molar ratios of NH$_3$(g) to particle-phase NH$_4^+$.



**Figure D2. Comparison of activity coefficients for different species.** (a)-(c) Comparison of activity coefficients involved in ISORROPIA A2 subcase calculations, as predicted by ISORROPIA and E-AIM. (d) Comparison of activity coefficients involved in ISORROPIA D3 subcase calculations, as predicted by ISORROPIA and E-AIM. (e) Mean activity coefficients predicted by ISORROPIA that are involved in ISORROPIA G5 subcase calculations.

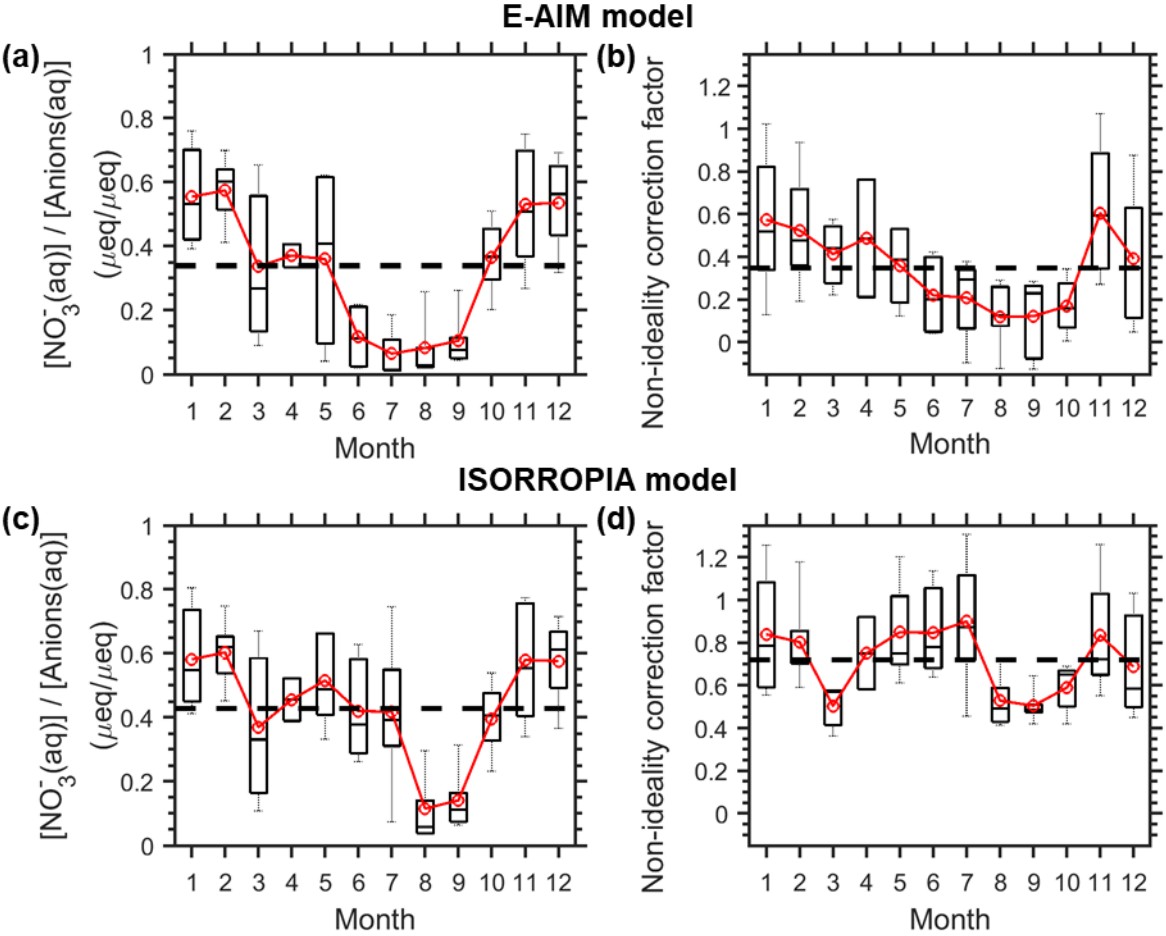


**Figure E1. Monthly variation of (a, c) NO₃- fraction in anions(aq), and (b, d) the corresponding non-ideality correction factors for Toronto site in 2012.** The data are estimated by (a-b) E-AIM model and (c-d) ISORROPIA model. The black dash lines represent the annual mean levels. The box and whiskers represent the 10[th], 25[th], 50[th], 75[th] and 90[th] percentiles, respectively, while the red markers represent the monthly means.