# Peer review of "Impact of non-ideality on reconstructing spatial and temporal variations of aerosol acidity with multiphase buffer theory"

_Atmospheric Chemistry and Physics, 2021_

## Author Comment (AC1)

**Manuscript No.**: acp-2021-55

**Title**: Impact of non-ideality on reconstructing spatial and temporal variations of aerosol acidity with multiphase buffer theory

5  We thank the referees for their valuable and constructive comments/suggestions on our manuscript. We have revised the manuscript accordingly and please find our point-to-point responses below.

**Comments by Anonymous Referee #1:**

*General Comments:*

10  *The pH values of aerosols are very important and attract lots of controversies, and are the hotspot in the investigation of aerosols properties. This article introduced a non-ideality correction factor and investigated its governing factors. Besides, a parameterization method was proposed to estimate cni at given RH, temperature and NO3- fraction, and was validated against long-term observations and global simulations. The results are very interest, and provides a way for pH retrieval when chemical*
15  *measurements are unavailable for the ammonia-buffered regions and periods. The manuscript is suitable to be published on Atmos. Chem. Phys. after considering the following comments.*

*Detailed Comments:*

*1) In the line 174-176, it is hard to understand the result that there was relatively small difference in pH*
20  *predictions by E-AIM and ISORROPIA, but higher difference in estimated cni. Can the authors provide some more detail information to explain this result?*

Responses:

We thank the reviewer for the comments. The smaller relative difference of pH than $c_{ni}$ predictions between E-AIM and ISORROPIA for the $NH_3$-$H_2SO_4$-$H_2O$ system (Fig. R1) is due to the ideal constant of $K_{a,NH3}^{*,i}$.
25  Based on the multiphase buffer theory and the definition of $c_{ni}$, we have:

$$\text{pH} = pK_{a,NH3}^{*,ni} + \log\frac{[NH_3(g)]}{[NH_4^+(aq)]} = pK_{a,NH3}^{*,i} + c_{ni} + \log\frac{[NH_3(g)]}{[NH_4^+(aq)]} \tag{1}$$

where the $pK_{a,NH3}^{*,i}$ is merely determined by AWC at fixed temperature. For the $NH_3$-$H_2SO_4$-$H_2O$ system, the E-AIM and ISORROPIA models generate similar prediction of AWC (and therefore similar ideal constant of $K_{a,NH3}^{*,i}$) and $[NH_3(g)]/[NH_4^+(aq)]$ (Fig. D1). Therefore, the absolute pH difference between these two models, $\Delta pH = pH_{E-AIM} - pH_{ISOR}$, is roughly equal to the difference of $c_{ni}$, i.e., $\Delta c_{ni} = c_{ni, E-AIM} - c_{ni, ISOR}$ (Fig. R1a). However, in terms of relative differences (defined as $|\Delta X|/X_{ave}$ here, where $X_{ave}$ refers to the average of X as predicted by ISORROPIA and that by E-AIM), we can see that:

$$\frac{|\Delta pH|}{pH_{ave}} \approx \frac{|\Delta c_{ni}|}{pK_{a,NH3}^{*,i} + \log\frac{[NH_3(g)]}{[NH_4^+(aq)]} + c_{ni, ave}} < \frac{|\Delta c_{ni}|}{c_{ni, ave}} \tag{2}$$

as $pH_{ave} > c_{ni, ave} > 0$ for the tested conditions (Fig. R1b,c). That is, the relative differences between these two models are generally smaller for pH predictions (<0.35; Fig. R1c) than $c_{ni}$ predictions (up to 7; Fig. R1b).

[Figure]

**Figure R1. Comparison of the differences of $c_{ni}$ and pH predictions between ISORROPIA and E-AIM models for the $NH_3$-$H_2SO_4$-$H_2O$ system. (a)** The differences in pH predictions between E-AIM and ISORROPIA ($\Delta pH$) against that of $c_{ni}$ ($\Delta c_{ni}$). **(b)(c)** The relative differences against average levels for (b) $c_{ni}$ and (c) pH, where the average levels **are** the averages of E-AIM and ISORROPIA predictions.

We noted that this statement can be confusing and is not closely related to the main idea of this part. Therefore, we've deleted this statement and revised the corresponding paragraphs into (see Line 178-190 in the revised manuscript):

"Although showing the same influencing factors, $c_{ni}$ estimated by E-AIM and ISORROPIA are not identical (Fig. 1). Especially for the $NH_3$-$H_2SO_4$-$H_2O$ system (i.e., $(NH_4)_2SO_4$ dominated aerosols), E-AIM

(Fig. 1a) and ISORROPIA (Fig. 1b) even predicted reversed trends in $c_{ni}$ dependence on RH and temperature. This is more clearly shown in Fig. 2 (blue dots), where $c_{ni}$ by E-AIM and ISORROPIA at the same conditions (i.e., same RH, temperature, and chemical profiles) are compared. As shown in Fig.2a, while $c_{ni}$ predicted by E-AIM ranged -0.3 to 0.5 for $(NH_4)_2SO_4$ dominated aerosols, that by ISORROPIA is

5    always larger than 0.1. This is mainly caused by the difference of calculated activity coefficients between ISORROPIA and E-AIM (Eq. 14b; see details in Appendix D, Figs. D1 and D2).

Despite the large difference in predicted $c_{ni}$ for the $NH_3$-$H_2SO_4$-$H_2O$ system, the E-AIM and ISORROPIA models generate similar prediction of AWC, and therefore similar ideal constant of $K_{a,NH3}^{*,i}$ (Fig. D1a). Combined with different $c_{ni}$, this would lead to different prediction of $[H^+(aq)][NH_3(g)]/[NH_4^+(aq)]$ by the

10    two models (Eq. 14c). However, with the constraint of charge balance and mass consevations of ammonia (Appendix D), the disagreement in the predicted molar ratios of $NH_3(g)/NH_4^+(aq)$ between these two models is relatively small (4%~6%; Fig. D1b), and most of the $c_{ni}$ variations is allocated to the $[H^+]$, or pH, predictions (Fig. D1c)."

15    *2) As the authors are mentioned, the cations of Na+, Ca2+, K+, and Mg2+ play a minor roles as their influence is more indirect. However, NH3/NH4+ plays important roles in multiphase buffer theory. What is the role of the NH3/NH4+ in the non-ideality coefficient?*

**Responses:**

When the non-ideality coefficient $c_{ni}$ changes, the model predicted $NH_3/NH_4^+$ will change resultantly.

20    Therefore, $NH_3/NH_4^+$ can reflect the $c_{ni}$ predictions and be used to derive $c_{ni}$. But, it is not the determinant of $c_{ni}$. The main factors that influence $c_{ni}$ are RH, temprature and the fraction of $NO_3^-$ in anions, as discussed in section 3.1 in the manuscript.

We've further clarified this point in the revised manuscript as (see Line 118-130 in the revised manuscript):

"We now define the non-ideality correction factor $c_{ni}$ to represent the difference of pH caused by non-

25    ideality. Based on Eqs. 8b and 13c, $c_{ni}$ is therefore:

$$c_{ni} = pK_{a,NH3}^{*,ni} - pK_{a,NH3}^{*,i} \tag{14a}$$

And combining Eqs. 13b and 14a, we have:

$$c_{ni} = -\log \frac{\gamma_{NH4+}}{\gamma_{H+}} \tag{14b}$$

Eq. 14b shows the intrinsic determining factors of $c_{ni}$, i.e., $\gamma_{NH4+}$ and $\gamma_{H+}$. Major influencing factors of $c_{ni}$ are therefore those influencing the activity coefficients (see section 3.1).

When $\gamma_{NH4+}$ and $\gamma_{H+}$ are not available, the $c_{ni}$ can be alternatively calculated by (Eqs. 13a, b):

(14c)

$$c_{ni} = pK_a^{*,ni} - pK_a^{*,i} = -\log(\frac{[NH_3(g)][H^+(aq)]}{[NH_4^+(aq)]}) + \log(K_{a,NH3}\frac{\rho_w}{H_{NH3}\ R\ T\ AWC})$$

5   Eq. 14c is valid as [NH$_3$], [NH$_4^+$] and [H$^+$] concentrations will vary as a result of changing $c_{ni}$. Note that while [NH$_3$]/[NH$_4^+$] and pH variations can relect the $c_{ni}$ variations and therefore be used to derive $c_{ni}$, they are not the determining factors of $c_{ni}$. As shown in Eq. 14b, $c_{ni}$ is determined by $\gamma_{NH4+}$ and $\gamma_{H+}$, which further depends mainly on RH, temperature and the fraction of NO$_3^-$ in anions (see section 3.1)."

---

## Author Comment (AC2)

**Manuscript No.**: acp-2021-55

**Title**: Impact of non-ideality on reconstructing spatial and temporal variations of aerosol acidity with multiphase buffer theory

5   We thank the referees for their valuable and constructive comments/suggestions on our manuscript. We have revised the manuscript accordingly and please find our point-to-point responses below.

**Comments by Referee Yunhong Zhang:**

*General Comments:*

10  *This paper is significant work to understand the total contribution of NH3 on the acid-base equilibrium of condensed phase of atmospheric particles. Especially for the case of concentrated aqueous phase at low RH, non-ideality correction factors are explored. This paper should be published with considering the two comments.*

*Detailed Comments:*

15   *(1) when it gives the definition of pKa,NH3\*i, physical significant of pKa,NH3\*i should be more clear if the authors provide more description, i.e., both condensed chemical compositions and NH3 content in gas phase determined the pH when chemical reactions in the particles change the pH of condensed phase, or other more better description easy understanding for readers.*

**Responses:**

20  Following the referee's suggestions, we've further clarified the meaning of $K_{a,NH3}^{*,i}$ as (see Line 87-93 in the revised manuscript):

"For typical ambient conditions when AWC varies between 1 to 1000 $\mu$g m$^{-3}$, the [NH$_3$(g)] is usually $10^5$ to $10^8$ times larger than [NH$_3$(aq)], and the above equation can be simplified into:

$$K_{a,NH3}^{*,i} = \frac{[\text{H}^+(\text{aq})][\text{NH}_3(\text{g})]}{[\text{NH}_4^+(\text{aq})]} = K_{a,NH3} \frac{\rho_w}{H_{NH3} \, R \, T \, \text{AWC}} \tag{1a}$$

25  And taking negative lognormal on both sides, we have pH is related to $pK_{a,NH3}^{*,i}$ (i.e., $-\log K_{a,NH3}^{*,i}$) as (Zheng et al., 2020):

$$pH = pK_{a,NH3}^{*,i} + \log \frac{[NH_3(g)]}{[NH_4^+(aq)]}$$

The multiphase buffer capacity of $NH_3/NH_4^+$ pair reached its local maximum when $pH = pK_{a,NH3}^{*,i}$, namely when $[NH_3(g)] = [NH_4^+(aq)]$. At given AWC and $T$, $K_{a,NH3}^{*,i}$ is constant."

5  *(2). line 80 in the equation, [NH3(g)] is equivalent molarity of gaseous NH3 in solution, its unit is molar.Kg-1. In this case Ka,NH3*,I= Ka,NH3(1+1/(HNH3RTAWC)), water density should not appear in the equation. The same is in 8a and 8b.*

**Responses:**

Many thanks. We double checked the related part and our original equations are correct. The confusion
10  may be potentially caused by the definition of AWC. In this study, AWC is represented in the unit of ($\mu$g water) / ($m^3$ air), instead of the water volume mixing ratio, $w_v$, of (L water)/(L air). With the current units applied, the term $H_{NH3}RT$ has the unit of unity, and $\rho_w$ is needed to convert the AWC to $w_v$.

To avoid such confusions, we've further clarified the units in the relevant equations as (see Line 80-86 in the revised manuscript):

15  "The multiphase effective acid dissociation constant of $NH_3$ under ideal conditions, $K_{a,NH3}^{*,i}$, depends only on AWC and temperature as (Zheng et al., 2020):

$$K_{a,NH3}^{*,i} = \frac{[H^+(aq)]([NH_3(aq)]+[NH_3(g)])}{[NH_4^+(aq)]} = K_{a,NH3}\left(1 + \frac{\rho_w}{H_{NH3}\,R\,T\,AWC}\right)$$

where AWC is in $\mu$g $m^{-3}$, and is mainly determined by air particulate matter concentrations and RH. The $\rho_w$ is water density in $\mu$g $m^{-3}$, and AWC/$\rho_w$ represents the aerosol water volume mixing ratio in the air in ($m^3$
20  water) / ($m^3$ air). [$NH_3$ (g)] represents equivalent molality (in mol $kg^{-1}$) of gaseous $NH_3$ in solution (see details in Zheng et al. (2020)). The $H_{NH3}$ is Henry's law constant of $NH_3$ in mol $L^{-1}$ $atm^{-1}$, $R$ is the gas constant of 0.08205 atm L $mol^{-1}$ $K^{-1}$, and $T$ is temperature in K."